# Cold Diffusion on the Replay Buffer:
# Learning to Plan from Known Good States

**Zidan Wang**[1]    **Takeru Oba**[2]    **Takuma Yoneda**[3]    **Rui Shen**[4]    **Matthew R. Walter**[3]    **Bradly Stadie**[1]
[1]Northwestern University    [2]Toyota Technological Institute
[3]Toyota Technological Institute at Chicago    [4]Yale University
`zidanwang2025@u.northwestern.edu, sd21502@toyota-ti.ac.jp, takuma@ttic.edu`
`rui.shen@yale.edu, mwalter@ttic.edu, stadiebradly@gmail.com`

**Abstract:** Learning from demonstrations (LfD) has successfully trained robots to exhibit remarkable generalization capabilities. However, many powerful imitation techniques do not prioritize the feasibility of the robot behaviors they generate. In this work, we explore the feasibility of plans produced by LfD. As in prior work, we employ a temporal diffusion model with fixed start and goal states to facilitate imitation through in-painting. Unlike previous studies, we apply cold diffusion to ensure the optimization process is directed through the agent's replay buffer of previously visited states. This routing approach increases the likelihood that the final trajectories will predominantly occupy the feasible region of the robot's state space. We test this method in simulated robotic environments with obstacles and observe a significant improvement in the agent's ability to avoid these obstacles during planning.

**Keywords:** Imitation Learning, Diffusion, Planning and Safety

## 1 Introduction

Consider a robot in a large, enclosed garden with a rock at its center. Humans initialize the robot's behaviors by collecting demonstration data of various tasks around the garden, such as pulling weeds, watering plants, and pruning hedges. During these demonstrations, humans naturally avoid the large rock, preferring to walk around it. At deployment, the robot successfully accomplishes the demonstrated tasks. However, with a slight change in its start location, the robot, attempting to generalize to this new situation, devises a plan that unfortunately sends it barreling straight into the rock—a classic case of faulty generalization. This scenario prompts the question: how should an agent determine which paths are suitable for generalization and which are not?

Recent techniques in LfD demonstrate significant potential by utilizing diffusion models to facilitate planning. In Diffuser [1], a robot's start and goal positions are pinned, and a temporal diffusion model fills in the rest of the trajectory, a process often referred to as in-painting. One major advantage of in-painting with diffusion is that it enables planning over an entire trajectory. Instead of auto-regressively considering one planned state at a time, in-painting allows agents to generate plans with temporal coherence across all planned states.

However, a notable limitation of Diffuser is that the algorithm's prior is a $d$-dimensional Gaussian space. In the context of imitation, this implies the algorithm begins with the assumption that all intermediate states between the start and goal can occur anywhere within a $d$-dimensional ball. This assumption becomes problematic if the robot's actual state space contains holes or infeasible regions (e.g., due to collisions). Although the diffusion process should, in theory, converge to a feasible trajectory that navigates around these areas according to the data distribution, we often find this not to be the case in practice. Small variations in the robot's start or goal positions frequently

7th Conference on Robot Learning (CoRL 2023), Atlanta, USA.

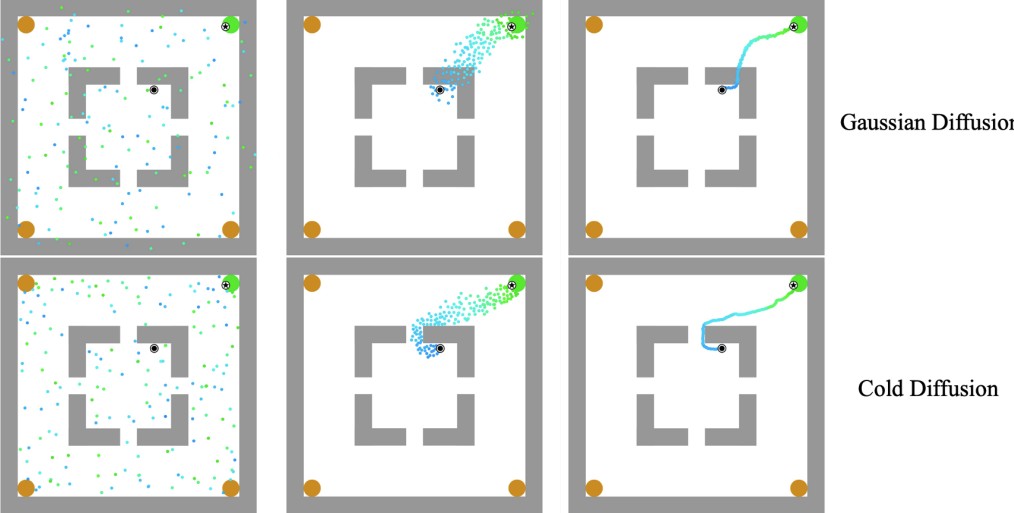

Figure 1: Cold Diffusion on the Replay Buffer: In contrast to (top) Gaussian diffusion, (bottom) we define a degradation operator that iteratively maps the points of a trajectory to randomly sampled points on the agent's replay buffer. We then train a restoration operator $\mathcal{R}$ to reverse this noising process. At test time, the agent randomly samples points from the replay buffer and reverses the diffusion process to obtain a trajectory that completes the desired task.

result in trajectories that attempt to traverse these infeasible regions, much like a garden robot might inadvertently crash into a rock.

In this paper, we explore the use of an agent's past experiences to guide the optimization of a diffusion model. Our proposed algorithm, Cold Diffusion on the Replay Buffer (CDRB), starts with a replay buffer filled with past trajectories. We iteratively apply a noising process that maps states along the trajectory to nearby points in the replay buffer. As this process repeats, the original trajectory degrades, transforming into a set of randomly distributed nodes across the replay buffer. Using cold diffusion, we train a restoration operator to reverse this noising process, mapping these noisy nodes back onto the original trajectory. During sampling, we achieve new goals by pinning a set of start and end positions and applying the restoration operator to generate a feasible path through the replay buffer.

In summary, our work makes significant contributions in the following ways:

- We introduce a novel cold diffusion model, CDRB, tailored for robotics environments. CDRB addresses challenges of diffusion-based planning methods that often generate infeasible plans.

- We demonstrate the effectiveness of our method in various planning tasks using simulated robotic environments with and without significant obstacles that may obstruct the robot's path.

## 2   Related Work

**Learning from Demonstrations (LfD)**   is a cornerstone of robotics [2, 3]. The density matching approach that we explore in this paper was pioneered by maximum entropy formulations of apprenticeship learning [4, 5, 6, 7]. GAIL [8] achieves density matching using a GAN-like discriminator [9] as a reward function. This formulation was subsequently extended to accommodate general f-divergence metrics [10, 11]. The constraint of this f-divergence optimization process has been further investigated [12, 13]. Meanwhile, others have demonstrated the utility of hindsight relabeling and Q-learning within the realm of imitation learning [14, 15]. Menda et al. [16] examined the concept of safe imitation learning by utilizing ensembles to correct learned policies mid-rollout,

as in DAgger [17]. Yin et al. [18] and Chen et al. [19] provide further insights into safe imitation learning, especially in the context of self-driving.

When it comes to learning action distribution, especially in continuous action space, the distributions have been often modeled as unimodal Gaussian. Since multiple "correct" actions may be possible in certain states, trying to model it with a unimodal distribution leads to a failure. For example, one can imagine a single state that shows up multiple times in the demonstration data is labeled with two very different actions. Trying to fit a Gaussian distribution will end up in placing its mean between those actions. To overcome this limitation, there have been several attempts to model multi-modal action distributions, with implicit functions [20], conditional VAE [21] or by discretizing action space[22]. On the other hand, diffusion models can naturally learn and generate a complex multi-modal distribution since it is designed to *iteratively* add or remove Gaussian noises. Given the success in computer vision community, people have recently started to explore their applications to control domains.

**Diffusion Models in Robotics** have enjoyed recent success, underscoring their potential and adaptability in addressing complex robotic tasks [23, 24, 25, 26, 27, 28, 29, 30, 31]. Diffuser [1] learns a diffusion process to generate entire trajectories conditioned on start and end state information. Diffuser primarily focuses on integrating Gaussian diffusion into the trajectory-planning framework in robotic settings, while we recognize the importance of designing a trajectory diffusion model that better aligns with the imitation learning context.

Pearce et al. [29] and Wang et al. [30] also apply diffusion models to control tasks, but they do so as state-conditioned action diffusion models, instead of trajectory diffusion models. In contrast to their approach, which looks at the problem from a policy optimization standpoint, we approach it through the lens of trajectory planning. Their framework takes in states and generates an action. Specifically, Pearce et al. [29] operates in analogy to other behavioral cloning algorithms, and Wang et al. [30] resembles the actor-critic algorithm, incorporating Q-value function guidance during training as the critic. In contrast, our approach can be viewed in analogy to multi-goal imitation learning, where the model learns to generate entire trajectories given specific start and goal states based on expert demonstration. As a result, both Pearce et al. [29] and Wang et al. [30] sample actions with real-time feedback, step by step, requiring continuous interactions with the environment, while our methodology has the capability to operate and generate complete plans without any real-time interactions during the sampling process.

Evaluation results show that Janner et al. [1], Pearce et al. [29], and Wang et al. [30] outperform state-of-the-art models across multiple decision-making tasks. Since they represent cutting-edge diffusion model applications in robotic environments, we adopt these methods as baselines in Section 5, providing a rigorous benchmark for our method.

Concurrent to our work, Xiao et al. [32] propose SafeDiffuser to ensure that diffusion probabilistic models satisfy specifications by using a class of control barrier functions. While the foundational motivation of SafeDiffuser resonates with ours, it relies heavily on predetermined safety parameters, such as specific subspaces that identify obstacles or traps. Yet, acquiring detailed and precise data of this nature is not always practical. Unlike SafeDiffuser, our method operates without predetermined safety specifics—we simply need expert trajectories. By design, these trajectories would not include any infeasible states, allowing our model to learn from actual experiences.

**Safe Planning and Exploration** is a perennial problem in the RL and control literature. There is a long history of research in the robotics community on sample-based motion planning that seeks to identify feasible trajectories in the presence of obstacles [33, 34, 35, 36, 37, 38, 39]. Work on the Bug Trap domain, where the agent has to escape a narrow corridor, was particularly inspiring for our current work [40]. Policy learning and exploration under constraints also trace their modern roots back to constrained MDPs [41]. Hans et al. [42] later crystallized the notion of learning that obeys feasible state constraints, though their approach requires human labels. Classic trajectory optimization literature at times maintains feasibility by recasting the problem as sequential convex

optimization with collision checking [43]. Modern work has sought to bridge modern RL and safe planning, often by optimizing stability [44, 45], reachability [46], or reversibility [47] to encourage safe behaviors. In the prior works that have delved into the challenge of generating collision-free trajectories [23, 24], their experiments heavily relied on predefined cost functions (i.e., energy functions). On a final note, a large number of works try to infer safety directly from human preferences [48, 49, 50, 51]. See Pecka and Svoboda [52] and García and Fernández [53] for comprehensive literature reviews of safety in RL.

## 3 Background

Learning from Demonstrations (LfD) is a general problem setting where a robotic system learns to copy a demonstrated behavior. Suppose we start with some demonstration trajectory $\tau = [s_0, a_0, s_1, a_1, \ldots, s_T, a_T]$. In the action-density matching variant of LfD, we assume there exists an underlying conditional action density $\rho_\tau(a_i|s_i, s_T)$. A policy $\pi$ is then trained to recover this distribution. In other words, we train $\pi$ to recover $\tau$'s conditional state-action distribution. Meanwhile, in the state-density matching variant of LfD, we train $\pi$ such that $D_{\text{KL}}(\rho_\pi(s \mid s_i, s_T) \parallel \rho_\tau(s \mid s_i, s_T))$ is minimized. In this variant, we only care about state-density matching, and action choice is optimized only to facilitate this matching.

The recently proposed Diffuser algorithm is an interesting case, because it considers the problem of density matching over an entire trajectory's state-action pairs. Given the initial $(s_0, a_0)$ and final $(s_T, a_T)$ state-action pairs, Diffuser attempts to recover all intermediate state-action pairs simultaneously. Let $\tilde{\tau} = [s_0, a_0, z_{1,s}, z_{1,a}, \ldots, s_T, a_T]$ be a trajectory with the desired start and end points, but the middle contains randomly sampled Gaussian noise. The Diffuser objective is to learn a mapping $R : \tilde{\tau} \to \tau$, minimizing the loss function $D_{\text{KL}}(\rho(R(\tilde{\tau})) \parallel \rho(\tau))$.

Given its name, it should come as no surprise that Diffuser accomplishes this density matching by leveraging diffusion models. Recently gaining popularity, diffusion models are generative models that learn to iteratively map some noisy input onto a target distribution. In the context of LfD, the model takes some noisy intermediate states $z_{1,s}, z_{1,a}, \ldots, z_{T-1,s}, z_{T-1,a}$ and iteratively denoises them to recover $s_1, a_1, \ldots, s_{T-1}, a_{T-1}$. Let us assume that we start with a noisy trajectory $\tau(t)$, where $t$ indexes the diffusion process timestep, not the temporal stage of the trajectory. The trajectory can be expressed as

$$\tau(t) = \alpha(t)\tau(0) + \sigma(t) \cdot z, z \sim \mathcal{N}(0, I), \tag{1}$$

where $\sigma(t)$ scales the noise magnitude at timestep $t$, and $\alpha(t)$ scales based on the data magnitude. The LfD problem is then to gradually remove the noise from $\tau(t)$ and recover $\tau(0)$. This can be done using a reverse stochastic differential equation that flows backwards from $t = 1$ to $t = 0$, at each step making use of a noise-perturbed score function $\nabla_\tau p_t(\tau)$. For example, if we assume Gaussian noise is added to the trajectory at each timestep, then the sampling procedure can be defined as

$$d\tau(t) = \left[ -\frac{d[\sigma^2(t)]}{dt} \nabla_x \log p_t(x) \right] dt + \sqrt{\frac{d[\sigma^2(t)]}{dt}} dw \tag{2}$$

with $dw$ being a Wiener process. Our goal is to learn the noise-perturbed score function $\nabla_\tau p_t(\tau)$. This can be done with denoising score matching. In particular, let $s_\theta(\tau(t), t)$ be the parameterized score model. The goal is then to learn the noise scale at time $t$

$$L_t = \mathbb{E}_{\tau(0) \sim \rho(\tau), z \sim N(0,I)} \left[ \|\sigma_t s_\theta(\tau(t), t) - z\|_2^2 \right], \tag{3}$$

where $\rho(\tau) = p_{\text{data}}$ is the demonstration density. During training, we consider a weighted sum over $L_t$, trying to learn the appropriate noise level across various points in time. We refer the reader to Ho et al. [54] and Song et al. [55] for more discussion on possible weighting strategies. At test time, new trajectories can be generated by fixing $(s_0, s_T)$, sampling $\tau(t) \sim \mathcal{N}(0, \sigma^2)$, and iteratively applying the Euler-Maruyama method with the learned $s_\theta(\tau(t), t)$ to recover $\tau(0)$

$$\tau(t) = \tau(t + \Delta t) + (\sigma^2(t) - \sigma^2(t + \Delta t))s_\theta(\tau(t), t) + \sqrt{\sigma^2(t) - \sigma^2(t + \Delta t)}z. \tag{4}$$

Note that $s_0$ and $s_T$ are held fixed during this process. This is referred to as *pinning*.

In contrast to diffusion, the recently proposed cold diffusion method [56] does not require a random degradation process. Instead, it learns a score function with respect to any degradation process $D(\tau(0), t)$ that transforms the data from $\rho(\tau(0))$ to $\rho(\tau(t))$. We write the cold diffusion procedure iteratively as

$$\tau(t-1) = \tau(t) - D(\tau(0), t) + D(\tau(0), t-1) \tag{5}$$

## 4  Methodology

We start by considering the Gaussian diffusion process in a simple 2D maze environment pictured in Figure 1. In this environment, the agent starts at some random position $s_0$ in the environment and must learn to navigate to one of the four target goal positions $s_T$ while avoiding the walls. The diffusion process samples noisy intermediate states from a Gaussian $\tau(t) = z_1, z_2, \ldots, z_{T-1}$, followed by reversing the noising process to obtain a planned trajectory $\tau(0) = s_1, s_2, s_3, \ldots, s_{T-1}$ connecting $s_0$ and $s_T$.

If the entire region connecting $s_0$ and $s_T$ is feasible, then the above diffusion process presents no issues. However, what happens when there is an infeasible region between the two points, for example a wall? Since the Gaussian score function at time $t$ is completely divorced from the realities of the environment, it is likely to sample intermediate points $z_i$ from the infeasible region, e.g., it will sample a point inside the wall. We hope that reversing the diffusion process will fix this issue and generate a feasible trajectory. However, we find in practice that the agent often tries to generalize incorrectly and generates a path that cuts through an infeasible region, especially when $s_0$ and $s_T$ were not present in the training data.

Perhaps the simplest fix to this infeasibility issue is by projecting the fully denoised trajectory $\tau(0)$ back onto the agent's feasible region. This can be accomplished with minimal overhead by keeping a replay buffer $\mathcal{B} = (\mathbf{s}, \mathbf{a})$ that contains all state-action pairs present in the expert demonstrations $\tau$. We then project each trajectory point $s_i(0), a_i(0) \in \tau(0)$ onto the closest replay buffer point

$$\arg\min_{s^*, a^* \in \mathcal{B}} \|s^* - s_i(0)\| + \|a^* - a_i(0)\| \tag{6}$$

Such a feasibility projection is intuitively satisfying. However, if we think about it, we will soon realize that such a projection will not work in our simple maze environment. The issue is that the entire planned trajectory attempts to go through the wall. Projecting this trajectory back onto the replay buffer will simply cluster all the infeasible points on either side of the wall. If the agent tries to follow this plan, it will naively go towards the wall and get stuck on one side, rather than correctly planning to avoid the wall entirely.

Nevertheless, we like the idea of this feasibility projection. Thus, we seek to incorporate it into the diffusion process itself, ensuring the entire optimization procedure is routed through the replay buffer. Specifically, we define a degradation process that consistently maps trajectory points $(s_i, a_i)$ to random points chosen from the replay buffer based on some distance schedule, denoted as $(s_i, a_i) \sim \mathcal{B}$. As $t$ increases, the maximum allowable distance between successive points $(s_i(t), a_i(t))$ and $(s_i(t+1), a_i(t+1))$ also increases. Cold diffusion can then be used to reverse this degradation process by starting from a random sample of points on the replay buffer $\tau(t) \sim \mathcal{B}$, and learning to iteratively map them back onto a trajectory $\tau(0)$ that starts at $s_0$ and reaches $s_T$. This procedure is described in detail below.

**Forward Process: Degradation**  We want to define a degradation process $D(\tau(0), t) = \tau(t)$ that takes points in a trajectory and iteratively spreads them out across the replay buffer. We want this degradation to vary continuously in $t$, with larger values of $t$ leading to sampled points that are more spread out across the buffer. We can consider this process in a pointwise fashion for each $s \in \tau$. Intuitively, at time $k$, we want to sample some point from the replay buffer that is within some small epsilon ball about $s$

$$p_k \sim B_{\epsilon_k}(s) \cap \mathcal{B} \tag{7}$$

We then replace point $s$ from the trajectory $\tau$ with the point $p_k$ from the replay buffer, $D(s_0, k) \rightarrow p_k$. As $k$ increases, $\epsilon_k$ will also increase, which will lead to more degradation as the sampled points stray farther from the original source $s$. We consider several schedules for increasing the size of $\epsilon_k$ in our experiments. For example, in a linear distance schedule, some max ball size is fixed based on the radius of the environment, $d_{\max}$. Then, $\epsilon_k$ varies linearly from 0 to $d_{\max}$

$$\epsilon_k = \frac{k}{t} \cdot d_{\max} \tag{8}$$

**Backward Process: Restoration**   The reverse process involves the restoration operator $\mathcal{R}$ that (approximately) inverts the degradation operator $D$. This operator has the property that

$$\mathcal{R}\left(\tau(t), t\right) \approx \tau(0) \tag{9}$$

In practice, we implement this operator as a neural network parameterized by $\theta$. We train the restoration network using the following objective

$$\min_\theta \; \mathbb{E}_{\tau \sim \rho(\tau)} \left\| \mathcal{R}_\theta(D(\tau, t), t) - \tau \right\|, \tag{10}$$

where $\tau$ denotes a random trajectory sampled from distribution $\rho_\tau$ and $\| \cdot \|$ denotes a norm, which we take to be $\ell_2$ in our experiments. This restoration process aims to recover the original trajectory from the degraded one. In particular, $D(\tau, t)$ degrades the trajectory to $\tau(t)$, and then $\mathcal{R}_\theta(\tau(t), t)$ attempts to reverse the degradation and recover $\tau(0) \approx \tau$. At test time, trajectories are generated by iteratively applying the restoration operator and then adding noise back to the states, with the level of added noise decreasing over time. Throughout this process, the start state $s_0$ and goal state $s_T$ remain pinned. Appendix A provides the full pseudocode of the training and sampling processes.

## 5   Experimental Results

We conducted experiments on a range of environments consisting of various challenging obstacles that prevent the agent from reaching target objects or goal locations in a trivial manner (i.e., straight line). We incorporated obstacles into existing environments to evaluate the agent's ability to navigate around obstacles and achieve its goals.

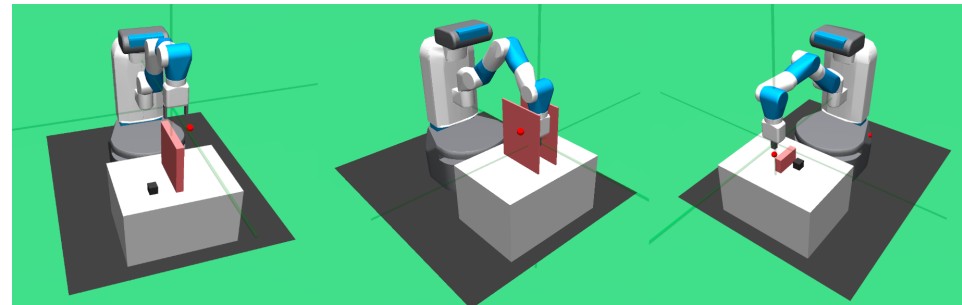

Figure 2: Environments considered in this paper. Left: Fetch robot must pick and place a box while avoiding a wall. Center: Robot starts between two walls and must reach a point outside the walls. Right: Robot must push a block to a desired point while avoiding an obstacle.

1) *Maze* and *Maze Tight*: These environments simulate a multi-goal maze with obstacles placed to divide the area at the center from the outside. There are four possible goals located at the corners of the environment, and one of them is randomly selected at each run. The Maze Tight variant is an adaptation of the original Maze, featuring narrower passages between the obstacles to test the algorithm's precise planning skills.

2) *Reach* and *Reach Obstacle*: In these environments, the primary objective is to navigate the robotic arm (gripper) to a specified target location. The Reach Obstacle variant introduces two tall walls on each side of the starting gripper position to test the robot's ability to reach the goal outside of the walls.

3) *Pick and Place* and *Pick and Place Obstacle*: These environments challenge the robot to grasp a block using the gripper and relocate it to a specified goal location. The Pick and Place Obstacle version of this environment adds further complexity by including a tall wall that the robot arm must go up or around to reach the block or goal location.

4) *Push* and *Push Obstacle*: In these environments, the robot's objective is to push a block to a target location without activating the gripper fingers. The Push Obstacle environment includes a short wall placed in the center.

For each environment, we first generated expert trajectories through the training of proficient agents via goal-conditioned policies. For the maze environments, multi-goal Soft Actor Critic (SAC) [57] models were used. For Reach, Pick and Place, Push, and their obstacle variants, we utilized Hindsight Replay Buffer (HER) [58] to train the agent. We compare against three recently proposed methods that leverage Gaussian diffusion for imitation learning: Diffusion-QL [30], Diffuser [1], and Diffusion BC [29]. We also included a baseline comparison where a multi-layer perceptron (MLP) is trained to concurrently predict entire trajectories, given specified start and goal states.

| Success Rate (%) | MLP | Diffuser | Diffusion BC | Diffusion-QL | CDRB (ours) |
|---|---|---|---|---|---|
| Maze | $74.22 \pm 3.81$ | $100.0 \pm 0.00$ | $90.13 \pm 1.21$ | $100.0 \pm 0.00$ | $95.33 \pm 2.05$ |
| Maze Tight | $38.29 \pm 1.80$ | $97.00 \pm 0.82$ | $89.21 \pm 1.20$ | $84.02 \pm 3.86$ | $98.00 \pm 0.00$ |
| Reach Obstacle | $80.74 \pm 5.20$ | $97.56 \pm 0.74$ | $89.63 \pm 0.41$ | $74.03 \pm 8.12$ | $97.92 \pm 0.18$ |
| Pick & Place Obstacle | $0.00 \pm 0.00$ | $46.68 \pm 1.29$ | $58.80 \pm 2.34$ | $24.19 \pm 8.31$ | $70.88 \pm 1.01$ |
| Push Obstacle | $2.43 \pm 1.10$ | $58.64 \pm 1.45$ | $34.81 \pm 7.79$ | $29.41 \pm 5.24$ | $71.04 \pm 1.58$ |
| Reach | $90.34 \pm 2.18$ | $100.0 \pm 0.00$ | $94.02 \pm 1.03$ | $58.60 \pm 14.69$ | $100.0 \pm 0.00$ |
| Push | $2.45 \pm 1.10$ | $70.56 \pm 0.68$ | $68.23 \pm 8.13$ | $65.60 \pm 12.63$ | $89.92 \pm 0.69$ |
| Pick & Place | $0.00 \pm 0.00$ | $92.36 \pm 1.21$ | $61.00 \pm 3.56$ | $23.60 \pm 9.63$ | $95.76 \pm 0.83$ |

Table 1: Benchmark results of test-time success rates goal reaching. Results over 5 seeds.

Table 1 shows the quantitative results over various planning methods across eight different domains. The mean and standard deviations are computed from trials over 5 different seeds. This shows that our approach consistently outperforms the baseline approaches in success rates except for Maze domain.

Qualitatively, we observe that Diffuser often generates an infeasible trajectory (e.g., some states are in the wall), while CDRB mostly generates a path in the feasible region. In addition, we also find that the plans generated by CDRB are often a lot smoother than those by Diffuser, as discussed in Appendix D.

Figure 3 shows an example of trajectories generated by Diffuser that directly cut through the wall and result in failures at execution time. In contrast, CDRB-generated trajectories would not cause the gripper to collide with the wall during execution.

## 6   Further Discussion

***k*-Means Clustering**   Keeping a full replay buffer can be costly, since diffusion across a larger buffer could have a higher propensity to make small adjustments at each reverse diffusion step. We thus experimented with applying $k$-Means clustering to reduce the buffer size. After clustering the buffer, we clip the resultant cluster centroids to the closest buffer points to ensure the feasibility of these states. The cluster centroids then become our new buffer. In general, we found this improved computational efficiency without degrading performance (see Figure 4(a)).

**Temporal Transformers**   Recent work has proposed the use of temporal transformers for diffusion-based imitation [29]. In contrast, the original Diffuser method uses a temporal U-Net [1].

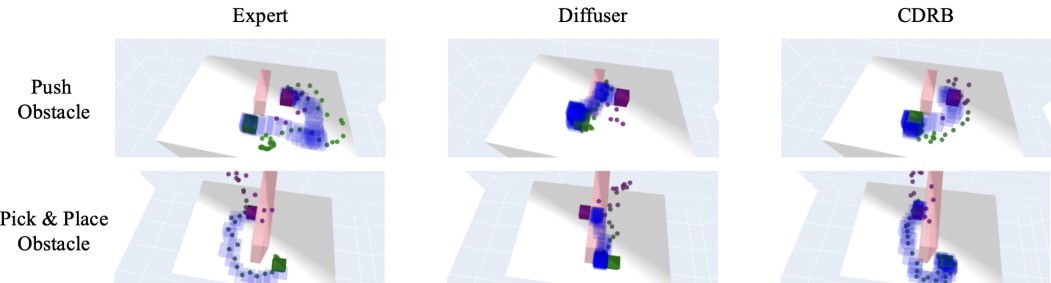

Figure 3: Trajectories on Pick & Place Obstacle and Push Obstacle. Diffuser-generated trajectories frequently struggle with crashing into the wall. In contrast, CDRB successfully learns to navigate around environmental wall constraints.

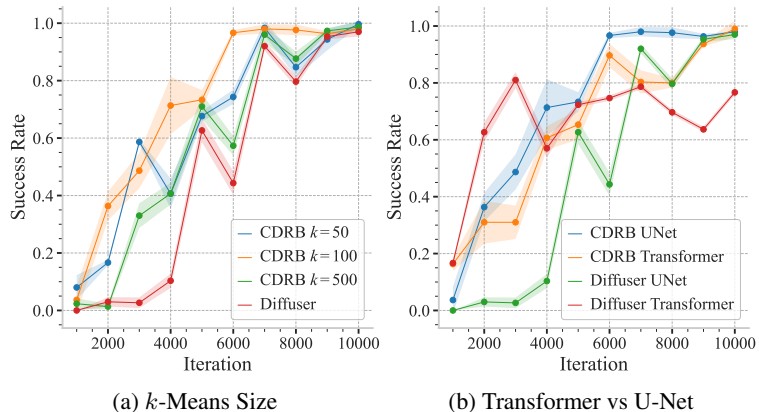

(a) $k$-Means Size

(b) Transformer vs U-Net

Figure 4: Comparative analysis of replay buffer efficiency via $k$-means clustering and performance dynamics of temporal architectures in diffusion processes.

Interested in the comparison, we explored both architectures for learning $\mathcal{R}_\theta$. Looking at Figure 4(b), we see that transformers generally converge much faster, but ultimately achieve worse final results. This is consistent with results in Table 1, where Pearce et al. [29] often perform slightly worse than Diffuser.

## 7 Limitations and Discussion

While we were generally impressed with the capabilities of CDBR, the algorithm has several bothersome limitations at this time. Among them, the algorithm seems sensitive to the number of reverse diffusion steps at test time. In fact, sensitivity in this regard is worse than most extant baselines. Because of our dependence on the replay buffer, the algorithm tends to struggle in areas of low visitation density, such as the Bug Trap environment that inspired much of our analysis [40].

Much like Diffuser [1], we find that directly using diffusion-generated actions often works poorly, and had to resort to inverse dynamics control. Yet, recent works have seen success with directly using diffusion for control [30, 31], which suggests it should be possible. In contrast to prior work [30, 31], we found our trained policies struggle to learn from noisy data. When initializing from human-provided demonstrations, rather than RL-based demonstrations, we found performance of all diffusion-based imitation learners suffered. This is a well-known problem in LfD [59]. We remain optimistic that future efforts in diffusion-based imitation will bear fruit in this area.

**Acknowledgments**

We would like to thank David Yunis for providing crucial advice on making the implementation of our forward diffusion process tremendously faster through vectorization. We also express our gratitude to our colleagues and reviewers for their invaluable feedback and insights that significantly improved this paper. This work was supported in part by the National Science Foundation under HDR TRIPODS (No. 2216899).

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

# A  Algorithm Pseudocode

---
**Algorithm 1** Training the Noise Model
---

**Require:** Expert trajectories $\mathcal{T}$, distance schedule $\epsilon$, number of diffusion steps $t$, replay buffer $\mathcal{B}$
**Output:** Trained restoration operator $\mathcal{R}_\theta$

1: repeat
2:    $\tau \sim \mathcal{T}$   (Sample a trajectory from expert trajectories)
3:    $k \sim (\{1, \ldots, t-1\})$   (Sample a diffusion step uniformly)
4:    for $i = 0, \ldots, T-1$
5:        $s_i(k) \sim B_{\epsilon_k}(s_i) \cap \mathcal{B}$   (Sample a RB point within a ball around $s_i$ with radius $\epsilon_t$)
6:    end for
7:    Construct noisy trajectory $\tau(k) = [s_0(k), s_1(k), \ldots, s_T(k), s_T(k)]$
8:    $\tau(0) = \mathcal{R}_\theta(\tau(k), k)$
9:    Perform gradient descent on $\min_\theta \mathbb{E}_{\tau \sim \rho(\tau)} \|\tau(0) - \tau\|$

10: until converged

---

---
**Algorithm 2** Sampling Process: Generating new Trajectories
---

**Input:** Replay buffer $\mathcal{B}$, start state $s_0$, goal state $s_T$
**Required:** Pretrained restoration operator $\mathcal{R}_\theta$, position controller $f_\phi$, number of diffusion timesteps $t$
**Output:** State-action sequence $\tau(0) = [s_0, a_0, \ldots, s_T, a_T]$

1:    Sample $T+1$ state-action pairs $\tau(t)$ uniformly from $\mathcal{B}$
2:    Pinning start and goal states: $s_0 \to \tau(t)[0], s_T \to \tau(t)[T]$
3:    for $k = t, t-1 \ldots, 1$ do
4:        $\hat{\tau}(0) = \mathcal{R}_\theta(\tau(k); k)$
5:        $\tau(k-1) = \mathcal{D}(\hat{\tau}(0); k-1)$
6:        Pinning start and goal states: $s_0 \to \tau(k-1)[0], s_T \to \tau(k-1)[T]$
7:    end for
8:    for $i = 0, \ldots, T-1$
9:        Compute the action $a_i = f_\phi(s_i, s_{i+1})$
10:    end for

---

# B  State Space Details

In our two maze environments, the state spaces are 4-dimensional, including $x$ and $y$ coordinates of the agent's position and its linear velocities in those directions.

The state space for Reach is 10-dimensional and consists of the gripper's kinematic information. It includes end effector positions and velocities in the $x$, $y$, and $z$ directions for the gripper, joint displacements, and velocities of the left and right gripper fingers.

The state spaces for Push and Pick and Place are 25-dimensional and consist of kinematic information for both the block object and gripper. Gripper dimensions include end effector positions and velocities in the $x$, $y$, and $z$ directions for the gripper, joint displacements, and velocities of the left and right gripper fingers. For the block, it involves positions, relative positions to the gripper, global rotation, relative linear velocity, and block angular velocity along the $x$, $y$, and $z$ axes.

## C   Experimental Details

In both the Maze and Maze Tight environments, we initially trained expert policies using the Soft Actor Critic (SAC) method for one million training timesteps, utilizing sparse reward [57]. Then, we collect the 8k demonstrations as the training dataset by using SAC, and train the diffusion models with this data. Since the lengths of the demonstration trajectories are different, we randomly crop the demonstration trajectories so that the trajectory length becomes equal for training. The cropped length is 168 in this experiment. In the inference stage, we randomly sample starts and goals while ensuring that the goal can be reached within 168 steps. The goals are sampled from 4 points in the center of the colored circle near the corner in Fig. 1. The starts are sampled from the inner region surrounded by 4 walls located around the center. Then, the diffusion model plans the trajectory by inpainting manner in the same way as diffusion planning [1]. The action is obtained by a position controller, which calculates the torque so that the agent follows the trajectory predicted by the diffusion model. If the agent touches the goal circle, the planning is regarded as a success.

In the Fetch environments, we obtained an expert policy with HER, trained for two million training steps. These environments were all multi-goal with sparse reward binary feedback. Goals are sampled uniformly from the environment, and involve reaching a target location or moving a block to the target location. We collected 5,000 demonstrations for each environment, with varying trajectory lengths. Specifically, in the Reach and Reach Obstacle environments, the trajectories were of length 30, while in the Pick And Place and Push environments, they were of length 40. For the Pick And Place Obstacle and Push Obstacle environments, the trajectories had a length of 60. It should be noted that on Pick And Place Obstacle and Push Obstacle, the best HER [58] controller we could train received only a 90% success rate, suggesting this environment is difficult even for RL agents. Like the maze environments, validation sets were used to measure the performance of diffusion models. Actions are obtained with a position controller, which allows the agent to follow the paths generated by the diffusion model. If the agent manages to get within $\epsilon$ of the target goal at the last timestep, the trajectory is counted as a success.

## D   Additional Ablation Analysis

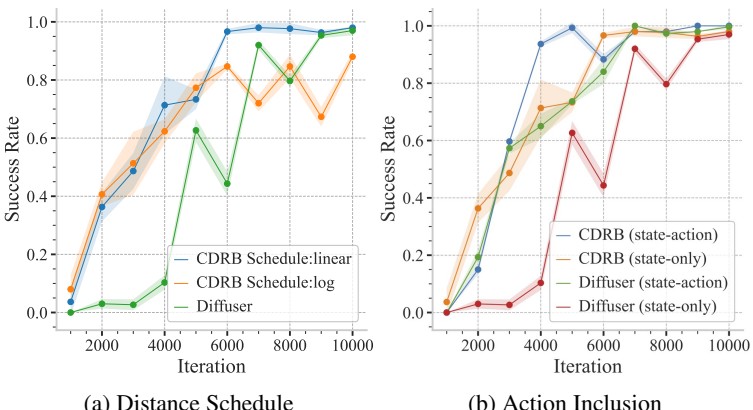

Figure 5: Comparative Analysis of Distance Schedules and Action Inclusion in Diffusion Processes.

**Distance Schedule:**  We wondered if changing the rate of degradation in our forward diffusion process would impact learning (Equation 7). Using both a log and a linear distance schedule, we see the log schedule learns slightly faster, but is generally less performant at convergence. See Figure 5 (a). We suspect a slower derogation schedule causes our algorithm to spend too much time optimizing smaller regions of the state space.

**Action Inclusion:** Given the use of an inverse dynamics model to generate actions, the use of actions in the diffusion target was completely optional. We conducted ablation experiments comparing state-

action to state-only sequences. Our findings indicate that CDRB's performance remains consistent, irrespective of action inclusion. In contrast, Diffuser's performance appears to be not only unstable when relying on states alone but also exhibits a slower learning curve across varying iterations. See Figure 5 (b).

**How smooth are the generated trajectories?** During evaluation, we noticed that trajectories generated by CDRB appeared smoother than those generated by Diffuser. See the example trajectories in Figure 6 (b) and (c). To test this, we divide the path length of the trajectories generated by Diffuser and CDRB by the path length of a trajectory generated by SAC. We see that under this metric, CDRB generates more efficient paths to the goal. In particular, it seems much better at locking onto a smooth region at the start of its trajectory, as opposed to Diffuser, which struggles at initial timesteps. This result is shown in Figure 6(a).

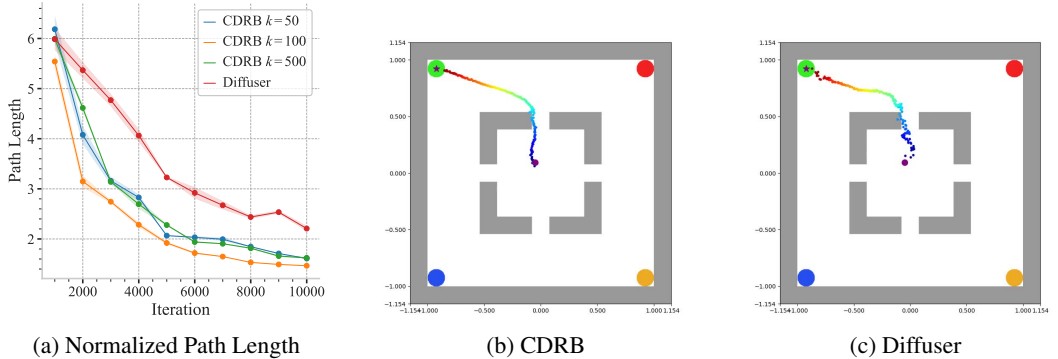

(a) Normalized Path Length        (b) CDRB        (c) Diffuser

Figure 6: Comparison of normalized path length and examples of the predicted trajectory by CDRB and Diffuser in Maze Tight environment.

