# OpenReview forum: "Cold Diffusion on the Replay Buffer: Learning to Plan from Known Good States"
_robot-learning.org/CoRL/2023/Conference — CoRL 2023 Poster_

### Official Review · Reviewer_Qj78 · 2023-07-16

**Confidence:** 3
**Originality:** Fair
**Technical Quality:** Fair
**Clarity Of Presentation:** Fair
**Impact:** 2

**Recommendation:**

Weak Reject: I recommend rejecting the paper, but will not argue for my recommendation if the majority of other reviewers have a different opinion.

**Review:**

In its current stage, I think that the paper is not written detailedly enough. Many points need to be clarified and some technical points need to be corrected. While the idea might be interesting, more work must be done to put this paper in a publishable state.

### Strengths
- interesting idea
- the paper is easy to read

### Weaknesses
- the paper is not well motivated
- while the paper is easy to read, it lacks detailedness
- some statements seem to be incorrect
- the generalization capabilities of the approach seem to be very limited
- many points need more clarification (see below)
- no real-world robotics experiments





**Quality Of The Limitations Section:**

Additional details required

**Questions For Rebuttal:**

### Main Points
- the restoration operator is not well described. In the introduction, the authors talked about a planner. In the methodology, they did not mention this at all.
- "In this work, we explore the problem of ensuring the feasibility of plans
generated by LfD" & "Because this entire diffusion process happens on the replay buffer, the resulting trajectory is guaranteed to be feasible." --> How can a trajectory generated with a trained inverse dynamics model be guaranteed to be feasible? Even with a perfect inverse dynamics model, how is the feasibility guaranteed when s_0 is not in the expert dataset?
-  this approach seems to have bad generalization capabilities as the target restoration operator's domain only includes states in the dataset. How well does it generalize in practice, especially when using only a few samples?
- this approach does not seem to scale well to complex higher dimensions as it needs to keep track of datasets. The proposed clustering solution is insufficient as it relies on a metric, which is difficult to get in all domains.
- why is the target restoration operator in the equation under line 179 only approximately equal to $\tau(0)$?
- why is it called cold diffusion on the "replay buffer" if, in the end, just a set of expert demonstrations are taken?

### Minor issues
- formulas don't have numbers
- line 116: correct sentence
- line 122: should not be $\tau(0)$
- line 132: Figure 1 is meant
- line 144: "ineffability issue"?
- supplementary should be in a separate file
- Algorithm 2 line 7: should be $s_{i+1}$ for the inverse dynamics model
- line 239: change -> changing

**Robotics Focus:**

Irrelevant to robotics

**Summary Of Paper:**

The authors propose a method to sample new trajectories using cold diffusion, given a set of expert demonstrations. They tackle the problem of wrong generalization apparent in diffusion models that rely on Gaussian noise for degradation by forcing the degradation process to go through the expert dataset. On a simple set of experiments, the authors show that the new degradation process results in better performance.

**Summary Of Recommendation:**

I ask the authors to clarify the above points and rewrite the paper to address the lack of clarity and technical correctness. Additional results would also be desired to clarify whether the mentioned generalization issue is evident in the low-data regime. If my points are addressed or some misunderstanding from my side is clarified, I am willing to raise my score, but in its current state, I vote for reject.

---

### Official Review · Reviewer_aADh · 2023-07-16

**Confidence:** 4
**Originality:** Fair
**Technical Quality:** Good
**Clarity Of Presentation:** Good
**Impact:** 3

**Recommendation:**

Weak Accept: I recommend accepting the paper, but will not argue for my recommendation if the majority of other reviewers have a different opinion.

**Review:**

Strengths:
* This paper presents a planning framework that uses cold diffusion to enable optimizations through the agent’s replay buffer of prior states.
* Figures are clear and helpful in understanding the method and results.

Weaknesses:
* The contributions of the paper are not clearly stated in the introduction.
* It needs to be clarified how this work is different from previous studies referenced in [1], [2], and [3].
* The paper’s proposed method seems to be based on the work of [1]. Specifically, it introduces the idea of using the agent’s past experiences to help guide a diffusion planner’s optimization. However, no direct comparison is made against the method proposed in [1] in the experiments section, making it difficult to assess the benefits of utilizing the agent’s prior experiences in the diffusion planner’s optimizations.
* No hardware experiments.

**Quality Of The Limitations Section:**

Limitations are addressed clearly

**Questions For Rebuttal:**

In this paper, my main concerns revolve around the identified weaknesses. Additionally, there are some important pieces of information that require clarification.
* 5.1 K-Means Clustering: how much computational efficiency is improved?
* Figure 4: reorder subplots to match the ordering of each topic in 5.1 Further Discussion.
* Figure 5: reorder subplots to match the ordering of each subplot mentioned in the text.
* Please formally introduce DDPM and Diffusion-QL.
* Please ensure all the references are up-to-date (e.g., [1] has been accepted to ICML 2022).
* What are the observation and action spaces in each environment?
* Why do the trained policies struggle in multi-modal environments even though diffusion models are known to excel at covering multi-modal distributions [2]?
* In the statement "Because this entire diffusion process happens on the replay buffer, the resulting trajectory is guaranteed to be feasible," how can we ensure that the resulting trajectory is feasible unless every resulting state comes from the replay buffer? If this is the case, it could be clarified in the paper.

Typos:
* Line 103 - “map” to “mapping”
* Line 239 - “change” to “changing”

**Robotics Focus:**

Highly relevant to robotics but no hardware experiments

**Summary Of Paper:**

The focus of this paper is on addressing the issue of plan feasibility by utilizing a temporal diffusion model. This model incorporates cold diffusion and draws on the agent's previous experiences to guide the optimization of a diffusion planner. A noising process is used to map states to nearby points in the replay buffer, and then cold diffusion is applied to reverse the noising process, ensuring that the resulting trajectory is feasible. Additionally, cold diffusion is used to address the sub-optimality of employing Gaussian diffusion when obstacles are present in the environment. The authors conducted experiments in various simulated environments where obstacles blocked the start and goal locations.


**Summary Of Recommendation:**

Based on the points raised and careful consideration, I suggest a weak reject. The main concerns are the missing clarifications on how this work differs from [1], [2], and [3] as well as the lack of a comparison with the method proposed in [1] when it is built on top of this work. Moreover, the list of issues in the questions for rebuttal should also be addressed.

[1] Janner et al, Planning with Diffusion for Flexible Behavior Synthesis, ICML 2022.

[2] Pearce et al, Imitating Human Behaviour with Diffusion Models, ICLR 2023.

[3] Wang et al, Diffusion Policies as An Expressive Policy Class for Offline Reinforcement Learning, ICLR 2023.


**Update:**
Based on the clarifications presented in the rebuttal, I believe they've addressed most of my concerns about this paper, and I am changing my recommendation to a 'weak accept.'

---

### Official Review · Reviewer_JCXJ · 2023-07-19

**Confidence:** 4
**Originality:** Good
**Technical Quality:** Good
**Clarity Of Presentation:** Excellent
**Impact:** 2

**Recommendation:**

Weak Accept: I recommend accepting the paper, but will not argue for my recommendation if the majority of other reviewers have a different opinion.

**Review:**

The paper is well written and it is easy to follow. The motivation is well presented and the contribution is clear. It might be interesting to highlight in bold the contribution of the work or maybe adding bullet points for even easier understanding of the work’s contribution.

The proposed approach is original, yet it is not evident that it will always work. On the one side, the degradation phase requires searching for the sample that is closest to an applied perturbation. This might be limiting when the dataset is too big. On the contrary, with a small dataset, there might not be sufficient datapoints in the vicinity to move.

Also, given the noise is added per state and not in the trajectory, there are no guarantees that the generated trajectory will be collision free. Particularly, close to the corners of the collision bodies or when the collision body is small.

The authors on the other hand are missing a proper citation. There exists some works that have explored the problem of generating collision-free trajectories [1,2], against which an experimental comparison would be important. It would be also nice to cite the works in the field of diffusion models for Robotics [3,4,5,6,7,8].

**Strenghts**

1. The problem of generating collision-free trajectories is important for Robotics. The authors proposes a novel and original approach to find trajectories that are collision-free.


**Weaknesses**

1. It is not fully evident the paper will guarantee collision-free trajectories. Given the constraint is set in the chosen states, it is possible to generate state transitions that goes through a collision. Even if two states are collision-free, the transitioning between two collision-free states might go through a collision. This situation is evident when the collision body is small or in the corners of the collision bodies.

2. The training algorithm seems to suffer with big datasets or too small datasets. Choosing a proper dataset might be hard in practise.

3. The authors are not presenting real robot experiments.

4. There exists previous works in the topic of generating collision-free trajectories with diffusion models [1,2]. It is essential to properly place the contribution of the work in contrast with previous works.



[1] Huang, Siyuan, et al. "Diffusion-based generation, optimization, and planning in 3d scenes." Proceedings of the IEEE/CVF Conference on Computer Vision and Pattern Recognition. 2023.

[2] Carvalho, Joao, et al. "Conditioned Score-Based Models for Learning Collision-Free Trajectory Generation." NeurIPS 2022 Workshop on Score-Based Methods. 2022.

[3] Liu, Weiyu, et al. "Structdiffusion: Object-centric diffusion for semantic rearrangement of novel objects." arXiv preprint arXiv:2211.04604 (2022).

[4] Urain, Julen, et al. "Se (3)-diffusionfields: Learning smooth cost functions for joint grasp and motion optimization through diffusion." 2023 IEEE International Conference on Robotics and Automation (ICRA). IEEE, 2023.

[5] Chi, Cheng, et al. "Diffusion policy: Visuomotor policy learning via action diffusion." arXiv preprint arXiv:2303.04137 (2023).

[6] Kapelyukh, Ivan, Vitalis Vosylius, and Edward Johns. "Dall-e-bot: Introducing web-scale diffusion models to robotics." IEEE Robotics and Automation Letters (2023).

**Quality Of The Limitations Section:**

Additional details required

**Questions For Rebuttal:**

The questions are directly connected with the highlighted weaknesses.

The most intriguing one is the one of the collision-free guarantees (Weaknesses 1). It is important the authors to clarify if they are able to guarantee collision-free trajectories and in case they are not, to explain how they could improve the method to obtain them.

**Robotics Focus:**

Highly relevant to robotics but no hardware experiments

**Summary Of Paper:**

This paper deals with the problem of generating feasible trajectories for collision-free robot navigation. The paper proposes a novel algorithm, called CDRB that learns a diffusion model that maps points along the dataset, in contrast with classical diffusion-based methods.

In CDRB, rather than adding white noise in the degradation phase, as it is common in Denoising score matching, they map the points to nearby points that exist in the dataset. Then, the network is trained to map the points back from the noisy point to the original trajectory.

In the experimental section, the authors compare their method against other baselines in collision-free trajectory generation.The authors show the benefit of their proposed method in contrast with baselines.

**Summary Of Recommendation:**

The paper presents an interesting idea to solve a robotics relevant problem.

Nevertheless, in the current format, I suggest a rejection. I suggest the authors to consider the weaknesses I highlighted to improve the paper.

---

### Official Review · Reviewer_MbuE · 2023-07-19

**Confidence:** 4
**Originality:** Good
**Technical Quality:** Good
**Clarity Of Presentation:** Very Good
**Impact:** 4

**Recommendation:**

Weak Accept: I recommend accepting the paper, but will not argue for my recommendation if the majority of other reviewers have a different opinion.

**Review:**

While the effectiveness of Cold Diffusion is still debatable in general application (https://openreview.net/forum?id=slHNW9yRie0), the authors have tried to justify why it makes more sense to use the data points in the dataset to degrade the data rather than adding gaussian noise. While, unlike diffusion models, there is no “theoretical” evidence or guarantee for such a phenomenon, the authors find cold diffusion to work better.

The authors refer Diffuser as DDPM across the paper, however, that is not the case. Diffuser incorporates classifier guidance by incorporating the expected return into the sampling process of obtaining the state-action sequence while keeping the start and goal state fixed via in-painting. Such a procedure is not yet established for cold diffusion.

If the authors plan to use an inverse dynamics model, why is state-action sequence sampled from the diffusion process? One can also sample state-only sequence (https://anuragajay.github.io/decision-diffuser/). The authors must include this as a baseline too in that case.

Further, recent works also show addition of safety in the reverse sampling process (https://arxiv.org/pdf/2306.00148.pdf). While the concern of the authors is justified that the samples might fall in the infeasible region, practically, several candidate solutions are sampled and the best one of them is chosen according to some metric (Q function, discounted return of the planned sequence). This weakens the overall motivation, however, the fact that cod diffusion is actually sampling efficiently is an important result that cannot be overlooked.

From the limitations mentioned by the authors, the proposed approach seems unstable in terms of several hyperparameters. Also, because only the provided data is used, it is definitely going to suffer from not modeling multi-modality as mentioned by the authors as well.

**Quality Of The Limitations Section:**

Limitations are addressed clearly

**Questions For Rebuttal:**

1. What is exactly the K-Means step? Does k=100 mean that you only have 100 states after clustering and clipping? If yes, then the degradation is done only using those selected 100 states? Because this can eventually mean overfitting to the low number of data.

2. Alg 2 in Appendix A in not complete, first it does not uses the sampling method mention in the paper and second, there is no inpainting of start and goal states. Appendix C is confusing without any proofs.

3. See Review Above

4. How will the proposed method behave if the start and goal states are out of distribution of the replay buffer? Like in Fig 5 (b), if replay buffer contains data from blue, green and yellow corners to goal in the center, and evaluated on a start state in red circle.

**Robotics Focus:**

Highly relevant to robotics but no hardware experiments

**Summary Of Paper:**

The paper implements “cold diffusion” for planning problems in robotics. The cold diffusion, unlike the standard diffusion process, is a Gaussian noise-free method that follows a degradation of the given data and learns a restoration model to reconstruct the original distribution. The authors employ a distance schedule to degrade a state in the trajectory by sampling a state in the dataset within a ball of scheduled radius around it. The planning formulation introduced by diffusion planning-Diffuser s followed to structure the trajectory as a sequence of (state, action) with given start and goal states. The authors show that the proposed method is better at handling infeasible regions or out-of-distribution areas and improves over prior diffusion model-based formulations. Further, implementation details have been discussed, and the choice of K-means clustering, UNet, and distance schedule has been justified. The results are presented in 4 simulated robotics environments.

**Summary Of Recommendation:**

The proposed method uses a recently introduced approach of cold diffusion to degrade the trajectory samples by using the data in the replay buffer and learns to restore the true trajectory from randomly sampled data from the replay buffer. Because everything happens inside the replay buffer, there is no possibility of generating datapoints in infeasible regions. This is a new direction in planning for robotics, and authors have shown success in several simulated robotics environments. However, there is also a problem that many papers have used Diffuser as a baseline but did not report this limitation. The motivation is somewhat questionable unless justified properly.

---

### Decision · Program_Chairs · 2023-08-30

**Decision:**

Accept (Poster)

**Comment:**

The paper presents a method to improve the feasibility of motion plans generated by conditional diffusion models. The method builds on the recently-developed cold diffusion technique and proposes to replace the gaussian noising process with a process that degrades an expert trajectory with randomly-selected samples in a replay buffer. The denoising process is thus to predict the original trajectory based on random samples from the replay buffer. Since the process happens over experience data, the generated trajectory is assumed (not guaranteed) to be feasible. The method is evaluated on a 2D maze environment and three simulated robotics tasks, with varying difficulties.

The reviewers have raised concerns regarding whether the method can generalize to out of distribution states and whether the generated plans are truly guaranteed to be collision-free. There are also concerns regarding missing reference to closely-related prior research and the lack of real-world experiment. At the same time, the reviewers found the problem of improving the feasibility of plans from a generative model to be well-motivated, and that the presented method to be interesting and sound.

Based on the reviews and the comments above, I recommend accepting the paper as a poster presentation. Please incorporate the writing and clarification suggestions in the next revision of the paper.